# Immune Checkpoint Inhibitory Therapy in Sarcomas: Is There Light at the End of the Tunnel?

**DOI:** 10.3390/cancers13020360

**Published:** 2021-01-19

**Authors:** Vasiliki Siozopoulou, Andreas Domen, Karen Zwaenepoel, Annelies Van Beeck, Evelien Smits, Patrick Pauwels, Elly Marcq

**Affiliations:** 1Department of Pathology, Antwerp University Hospital, 2650 Edegem, Belgium; Karen.Zwaenepoel@uza.be (K.Z.); patrick.pauwels@uza.be (P.P.); 2Center for Oncological Research (CORE), Integrated Personalized and Precision Oncology Network (IPPON), University of Antwerp, 2610 Wilrijk, Belgium; Andreas.Domen@uantwerpen.be (A.D.); evelien.smits@uantwerpen.be (E.S.); Elly.Marcq@uantwerpen.be (E.M.); 3Department of Oncology, Antwerp University Hospital, 2650 Edegem, Belgium; 4Department of Orthopedics, Antwerp University Hospital, 2650 Edegem, Belgium; Annelies.VanBeeck@uza.be; 5Center for Cell Therapy and Regenerative Medicine, Antwerp University Hospital, 2650 Edegem, Belgium

**Keywords:** immunotherapy, immune checkpoint inhibitors (ICIs), immune checkpoint blockade (ICB), sarcoma, programmed death-ligand 1 (PD-L1), predictive biomarkers, clinical trials

## Abstract

**Simple Summary:**

Soft tissue and bone sarcomas is a very heterogeneous group of tumors that has an aggressive course, especially in the metastatic setting. In this group the therapeutic options are rather limited. Immunotherapy is nowadays used successfully for the treatment of various tumor types. However in sarcomas this is still not the case. In this review article we aim to present all the available published information from clinical trials about the results of using immune checkpoint blockade as a therapeutic agent in sarcomas. Moreover, we try to unravel the possible prognostic biomarkers that may play here an important role.

**Abstract:**

Soft tissue and bone sarcomas are a very heterogeneous group of tumors with many subtypes for which diagnosis and treatment remains a very challenging task. On top of that, the treatment choices are limited, and the prognosis of aggressive sarcomas remains poor. Immune checkpoint inhibitors (ICIs) have drawn a lot of attention last years because of their promising response rates and their durable effects. ICIs are currently widely used in the daily routine practice for the treatment of a different malignancies, such as melanoma, Hodgkin lymphoma, and non-small cell lung carcinoma. Still, ICIs are not included in the standard treatment protocols of the different sarcoma types. However, a plethora of clinical trials investigates the clinical benefit of ICIs in sarcomas. There is clear need to develop predictive biomarkers to determine which sarcoma patients are most likely to benefit from immune checkpoint blockade. This review will focus on (i) the clinical trial results on the use of ICIs in different sarcoma types; and on (ii) possible biomarkers predictive for the effectiveness of these drugs in sarcomas.

## 1. Introduction

### 1.1. Sarcomas

Soft tissue and bone sarcomas are very rare neoplasms and account for less than 1% [1,2,3] of all malignancies. Although we refer to mesenchymal tumors as an entity, there are more than 200 distinct categories recognized and described in the latest World Health Organization classification of tumors book [4]. The rarity of these tumors leads in to difficulties defining the right criteria for diagnosis and precise treatment. Moreover, the heterogeneity makes prognosis difficult to assess. Many different parameters have been investigated in order to establish prognostic criteria for sarcomas. Among them, the tumor grade has been proven to be one of the best predictors of metastatic risk and progression free survival [5].

The etiology of most sarcomas remains unknown; however, genetic events have been attributed as being the main cause of mesenchymal tumorigenesis. According to their genetic alterations, sarcomas can be subdivided in two main categories. In the first category, sarcomas display a simple karyotype which can be a somatic mutation, a gene translocation or amplification that represents the driver oncogenic mechanism of the tumor. The second category includes sarcomas with a complex karyotype showing multiple aberrant chromosomal alterations [6,7,8] and represents almost two thirds of the sarcomas.

Some of the molecular events found in sarcomas are druggable, such as tyrosine-protein kinase (KIT) mutations in gastrointestinal stromal tumors (GISTs) and in a minority of other mesenchymal tumors. Unfortunately, today most of the oncogenic driver alterations are undruggable.

Surgery remains the golden standard for the treatment of localized disease. In case of larger tumors that cannot be resected completely, adjuvant radiation therapy can be applied to control the local aggressiveness. Unresectable sarcomas can also be treated with radiotherapy and/or chemotherapy in neoadjuvant setting. Chemotherapy is used in specific subtypes, such as rhabdomyosarcoma, osteosarcoma, and Ewing sarcoma. However, effective treatment of advanced sarcoma remains a challenge. Moreover, the five-year survival rates for the metastatic setting does not exceed 16%, thereby highlighting the need for new therapeutic strategies in sarcoma.

### 1.2. Immunotherapy

Immune checkpoint blockade (ICB) (referred also as immune checkpoint inhibitors—ICIs) is a well-known immunotherapeutic approach widely used due to the promising results in several cancer types. Inhibitory immune checkpoints (ICs) are responsible for controlling and inactivating the immune system in order to avoid autoimmunity. ICs are expressed under normal physiological conditions by different immune cell types [9]. Unfortunately, tumor cells can hijack this system. This results in T-cell exhaustion, immune tolerance and eventually suppression of the anti-tumor immune response. By blocking ICs, silenced anti-tumor responses will be reactivated [10,11].

A broad range of different ICs has been identified to date. One of the most commonly known is programmed death-1 (PD-1) expressed a.o. on T-cells. It can bind to programmed death ligand-1 (PD-L1) expressed on tumor cells and other cells. Today several drugs have been developed that can block the interaction between PD-L1 and PD-1, thereby reactivating silenced immune responses. The PD-1 and PD-L1 blockers that are widely used in the clinical practice are nivolumab, pembrolizumab, atezolizumab, durvalumab, avelumab, and cemiplimab, while many new ones are tested in clinicals trials. Another example of an IC is cytotoxic T-lymphocyte-associated antigen4 (CTLA-4) which binds with B7 on the antigen presenting cells (APC) [12]. In this context, ipilimumab is a widely known CTLA-4 blocking antibody that has been extensively used for the treatment of metastatic melanoma [13]. Ipilimumab in combination with nivolumab has also been approved as first-line treatment for advanced renal cell carcinoma and non-small cell lung cancer [14,15].

Although we will focus on ICB in this review, it is important to mention that the term immunotherapy refers to a broad range of different therapies. According to the National Cancer Institute (NCI), there are five main categories within immunotherapy: (1) T-cell transfer therapy, (2) monoclonal antibodies, (3) cancer treatment vaccines, (4) immune system modulators, and (5) ICB [16]. In addition to these categories, there are also new generation immune checkpoints with stimulatory effect, or checkpoints concerning innate immunity or natural killer cells, which gain more and more research interest in the field of immunotherapy [17].

Despite the rapid developments in the field of immunotherapy the last decade, ICB is not included in the standard treatment protocols of sarcomas. However, immunotherapy is currently under investigation in several clinical trials that include sarcomas. So far, no effective immunotherapeutic strategy for sarcoma has been identified. Given that ICB is a widely used immune therapeutic strategy in daily clinical practice together with the imperative need for new therapeutic alternatives for sarcomas, this review will (i) outline the results of clinical trials on the application of ICB in sarcomas, and (ii) discuss the possible mechanisms why this treatment has not been proven effective for sarcomas at present.

## 2. Online Searching Method

### 2.1. ClinicalTrials.gov

For this review we started with an extensive search on ClinicalTrials.gov for clinical trials that describe results in the application of ICI therapy on different types of soft tissue and bone tumors, until 11 November 2020. For our primary search we used the following three search terms: the first term was “sarcoma” AND the second term was variable, being “immune check point inhibitory therapy” (4 results) OR “immune checkpoint inhibitory therapy” (4 results) OR “immune checkpoint blockade” (0 results) OR “CTLA-4” (33 results) OR “anti-CTLA4” (14 results) OR “PD-L1” (62 results) OR “anti-PD-L1” (14 results) OR “PD-1” (70 results) OR “anti-PD1” (31 results). We also checked other checkpoint inhibitors such as “VISTA” (0 results), “LAG-3” (5 results) and “TIM-3” (4 results) and saw that these 9 studies were already covered by our broad primary search. Moreover, as second term we also used the approved ICI that are used in the daily routine practice at the moment, being “ipilimumab” (28 results) OR “nivolumab” (43 results) OR “pembrolizumab” (34 results) OR “atezolizumab” (13 results) OR “avelumab” (9 results) OR “durvalumab” (15 results) OR “cemiplimab” (2 results). When comparing all outcomes from the second searching term, there was considerable overlap of the clinical trials and we finally ended up with 149 different clinicals trials in sarcomas treated with ICI therapies. As a third search term we used “study with results” in the pool of 149 clinical trials and only 5 clinical trials were left.

### 2.2. Pubmed

The number with results of clinical trials with results on ClinicalTrials.gov seemed very low. Therefore, we investigated each of the 149 clinical trials in Pubmed by NCT number for any published results and we found publications for 26 of our NCT numbers. Three out of those 26 numbers overlap the 5 clinical trials reported with results on ClinicalTrials.gov, resulting in 28 clinical trials with results.

### 2.3. Summary

After thoroughly searching, only 14 out of these 28 clinical trials with results met the searching terms sarcoma AND any of the ICI listed above AND study with results on ClinicalTrials.gov and/or Pubmed. Table 1 summarizes these 14 clinical trials and the results in terms of the primary endpoint, while Table 2 gives an overview of all the predictive biomarkers that are investigated in each one of those clinical trials.

## 3. Results

### 3.1. Immune Checkpoint Inhibitors as Monotherapy

One of the largest studies is the SARC028 multi-institutional phase II study that assessed the safety and activity of pembrolizumab in patients with advanced sarcoma (NCT02301039) [18]. The trial included 80 patients divided equally in two treatment groups, one including patients with soft tissue and one including patients with bone sarcoma. In the soft tissue category patients with leiomyosarcoma (LMS), poorly differentiated/dedifferentiated liposarcoma (DDLPS), undifferentiated pleiomorphic sarcoma (UPS)/malignant fibrous histiocytoma, and synovial sarcoma were enrolled. The bone tumor category consisted of osteosarcoma, chondrosarcoma, and Ewing sarcoma. Pembrolizumab was used as monotherapy and was administered at 200 mg intravenously every three weeks for both groups, until disease progression or unacceptable toxicity.

Primary outcome of the study was investigator-assessed objective response (OR) according to response evaluation criteria in solid tumors (RECIST) version 1.1. This was defined as the proportion of patients in each cohort with a best overall response of complete (CR) or partial (PR) response. The trial did not reach the prespecified OR of 25%. Seven of 40 patients (17.5%) with soft tissue sarcoma achieved an OR. The median progression free survival (PFS) was 18 weeks. The 12-week PFS was 55%, suggesting clinical activity for soft tissue sarcomas with the majority being UPSs and DDLPSs. For UPS, the median duration of response was 49 weeks, suggesting that ICIs can have durable effects, especially for this group. Only one patient with synovial sarcoma showed a short-lived PR, while no patient with leiomyosarcoma displayed OR.

Confirmed PR was observed in 2 of 40 bone sarcomas showing a substantial shrinkage of tumor volume and a durable effect of more than six months. Median PFS was eight weeks. Anemia and decreased lymphocyte count were the most persistent toxic events often resulting in grade 3 or worse toxicity. The investigators tried to find out if there is a correlation between immunohistochemical (IHC) expression of PD-L1 and response to therapy. Three samples, all from UPS patients, had a PD-L1 expression in more than 1% of the tumor cells, still in general no statistical correlation could be shown. The investigators concluded that ICIs induced durable responses and showed meaningful clinical activity in patients with soft tissue sarcoma, in particular UPS and DDLPS.

The ADVL1412 (NCT02304458) [19] study is another multicentric, single arm, phase I–II trial investigating the safety, pharmacokinetics, and anti-tumor activity of nivolumab as monotherapy in children and young adults with recurrent or therapy refractory tumors, including soft tissue and bone sarcomas (particularly rhabdomyosarcoma, Ewing and osteosarcomas). The study had several primary objectives: (1) to determine the tolerability and describe the toxicity of nivolumab at the adult recommended dose; (2) to determine the systemic exposure of nivolumab in children compared to that in adults; (3) to determine the maximum tolerated dose in children; and (4) to explore the anti-tumor activity of nivolumab in selected childhood solid tumors or lymphoma. Secondary objectives included investigating the presence of infiltrating lymphocytes and PD-L1 expression in tumor specimens from patients.

Nivolumab 3 mg/kg was well tolerated and confirmed as the pediatric recommended phase II dose. To investigate dose-expansion the primary outcomes were tolerability, systemic exposure, maximum tolerated dose, and anti-tumor activity of nivolumab at the adult recommended dose in children and young adults.

A higher frequency of hematological toxicity was found in children but in general, administration of 3 mg/kg every 14 days was well tolerated. The most common immune related adverse events (irAEs) were increased lipase levels and cardiac and pleural effusion. No OR was observed and nivolumab as monotherapy did not show significant anti-tumor activity in the pediatric tumors. The study showed a low PD-L1 expression on the different sarcoma types and a paucity of infiltrating T-cells, emphasizing the possibility that other factors/mechanisms, such as tumor mutational burden (TMB), might play a role in the response.

Nivolumab as single agent was also investigated in a single-center phase II trial in patients with uterine LMS (NCT02428192) [20]. Twelve patients with metastatic or unresectable disease that were previously treated with chemotherapy were included. The primary endpoint was objective response rate (ORR). Secondary, they investigated the correlation between response to nivolumab and PD-1, PD-L1 and PD-L2 expression on available tissue samples. The patients received 3 mg/kg intravenous nivolumab every two weeks. None of the 12 patients experienced OR. The overall median PFS was 1.8 months while the median overall survival (OS) was not met. Clear correlation with IHC expression of PD-1, PD-L1 and PD-L2 could not be documented. Nine of 12 patients had grade 3 AEs or higher, with an increase of serum amylase and lipase being correlated to the drug administration.

Another humanized anti-PD1 antibody, geptanolimab, was investigated in Gxplore-005 phase II study (NCT03623581) [21] in adult patients with unresectable, recurrent or metastatic alveolar soft part sarcoma (ASPS) and provides evidence that suggest that immunotherapy could play an important role in the treatment of ASPS. Primary end point was ORR. Thirty-seven patients were included, which received 3 mg/kg intravenous geptanolimab every 2 weeks until disease progression or significant toxicity occurred. OR was 37.8%, which is significantly higher than the 10% expected from chemotherapy [22]. Limited patients developed grade 3 AEs, such as anemia, fever, and hypophisitis. In this study the expression of different biomarkers, such as PD-L1 status, microsatellite instability (MSI), TMB and immune infiltration with CD4^+^ cells, was compared with response to therapy. About 30% of the patients had PD-L1 combined positive score (CPS) of ≥ 1 in tissue samples; however, there was no difference in response to ICI therapy among positive and negative patients. The percentage of baseline CD4^+^ T-cells was significantly higher in non-responders than in responders, indicating that non-responding tumors may be rich in regulatory T-cells (T-regs) that suppress immune response. No MSI was present and all samples showed very low TMB.

ASPS was also a field of interest of the phase II clinical trial from Japan. The OSCAR study investigated the possible role of nivolumab in the treatment of advanced clear cell sarcoma (CCS) and ASPS [23]. The trial presented its preliminary results in CTOS 2020. Eleven CCS and 14 ASPS patients received nivolumab 240 mg every two weeks until disease progression or intolerable drug toxicity. The primary endpoint, which was response rate (RR), was not met. Nevertheless, encouraging was that the disease control rate reached 64% for unresectable CCS and ASPS. Median PFS was 4.9 months and median OS was 15.8 months.

The NCI 08-C-0007 (NCT01445379) [24] phase I study investigated the safety and pharmacokinetics of ipilimumab. Thirty-one patients between 2 and 21 years old with refractory or recurrent solid tumors were included. Seventeen of them had sarcoma of various histological types. The primary endpoint was to determine the tolerance and toxicity of ipilimumab in the young population. Secondarily, the trial aimed to quantify the anti-tumor effects of ipilimumab in the target group. Patients received ipilimumab with a dose escalation from 1 mg/kg up to 10 mg/kg. Grade 3 and 4 irAEs were seen with 5 mg/kg and 10 mg/kg dose, mostly related to gastrointestinal and liver toxicity. Because an increase in irAEs was seen in children under 12 years old treated with 10 mg/kg, the cohort of this age group was expanded. Interestingly, OS was better in patients with irAEs than in patients without irAEs, suggesting that irAE is an undesirable side effect of a desired result, i.e., the activation of the immune response. The investigators concluded that given the toxicities and inability to predict toxicity or response, ipilimumab as single agent in the pediatric tumors has no leading role.

The investigators of the nonrandomized phase I study (NCT02595866) [25] had a unique scope. They focused on safety of administrating pembrolizumab in human immunodeficiency virus (HIV)-positive patients who developed diverse non-HIV and HIV related malignancies, such as Kaposi sarcoma (KS). The secondary endpoint was to evaluate the tumor response. The study included 30 patients, 6 of which had KS.

In general, irAEs were similar to those described in non-HIV patients that received ICI for any other FDA-approved indications. Five out of six KS patients demonstrated tumor regression, yet did not met the criteria for PR. One patient with pretreatment KS-herpesvirus viremia died through a polyclonal KS-herpesvirus-associated B-cell lymphoproliferative disease.

The investigators concluded that ICIs can be safely administrated in patients with HIV, but caution for patients with active viremia.

Response to pembrolizumab was investigated also in patients with endemic and classic type KS in a prospective phase II clinical trial (NCT03469804) [26], published its results at the ESMO 2020. The study included 17 patients, which were given pembrolizumab 200 mg intravenously every 3 weeks for up to 6 months. The primary endpoint was best ORR. Almost 71% of the patients experienced an OR, while another 24% had SD. The irAE were tolerable with only one grade 3. A key finding of the study is that patients with lack of PD-L1 expression on tumor and immune cells on baseline tissue samples, had a limited effect with pembrolizumab treatment. This raises the question whether PD-L1 could become a predictive factor for response of endemic/classic type KS to ICB.

The AcSé study is a non-randomized, phase II clinical trial (NCT03012620) that investigated the response of pembrolizumab on different sarcoma histologic subtypes and demonstrated its results at the ESMO 2020 [27]. Twenty-four of the included patients had chordoma, 13 had ASPS, 6 had desmoplastic small round cell tumor (DSRCT), another 6 smarca4-malignant rhabdoid tumor (SMRT), and 31 had other histologic subtypes. The patients received pembrolizumab 200 mg intravenously every three weeks for up to two years. Best response was PR in 16% and SD in 36%. The investigators of this study highlighted the importance of histological type in response to treatment, as 50% of responses were observed in SMRT and 39% in ASPS patients.

### 3.2. Immune Checkpoint Inhibitors as Combination Therapy

The Alliance A091401 (NCT02500797) [28] is a randomized phase II trial investigating nivolumab with or without ipilimumab in patients with metastatic or unresectable sarcoma who received at least one previous line of systemic therapy. Nivolumab 3 mg/kg was given every two weeks or nivolumab 3 mg/kg and ipilimumab 1 mg/kg every three weeks until disease progression or up to two years after registration.

The primary endpoint was confirmed OR defined as CR and PR by RECIST version 1.1. Secondary endpoints were duration of response, the proportion of patients achieving a clinical benefit, PFS and OS.

All patients receiving treatment experienced irAEs. The most common grade 3 or worse irAEs in both cohorts were anemia and increased serum lipase levels. Monotherapy was generally tolerated better compared to the combination therapy. Two patients of the monotherapy group had confirmed PR, one with ASPS and one with non-uterine LMS, resulting in RR of 5%. The RR in the combination group was higher, reaching 16%. The median PFS was 1.7 months with monotherapy versus 4.1 months with the combination therapy. The clinical benefit of nivolumab monotherapy was not equal when compared to the currently available treatment options [29]. Moreover nivolumab monotherapy did not meet the predefined primary endpoint.

The combination therapy met its predefined primary endpoint, with median OS of 14.3 months while OS described for similar patient populations treated with selective tyrosine kinase inhibitor (TKI) is approximately 11–15 months [30]. These findings suggest that nivolumab as a single agent may not be active, and that only the combination therapy shows efficacy that may justify further studies as a treatment option for metastatic sarcoma patients.

A prospective, phase II clinical trial (SWONG S1609, cohort 51), that presented its results at the ASCO 2020 (NCT02834013) [31], investigated the combination of ipilimumab 1 mg/kg every six weeks and nivolumab 240 mg every two weeks, both administrated intravenously, in patients with metastatic or unresectable angiosarcoma. The primary endpoint was ORR, while the secondary endpoints were multiple, including PFS and OS. There were nine cutaneous and seven non-cutaneous angiosarcomas. ORR was 25% and six-month PFS was 38% regardless the primary localization, while 60% of the cutaneous angiosarcomas had a confirmed OR. The investigators made here also a comment about UV light exposure DNA mutational signature in cutaneous angiosarcomas, implying that this may interfere with the drug efficacy.

### 3.3. Immune Checkpoint Inhibitors Combined with Chemotherapy

Chemotherapy, particularly metronomic cyclophosphamide (MC), has been described to show immunological properties by depleting regulatory cells and restoring T- and natural killer- (NK) effector factors in cancer patients [32]. The PEMBROSARC was a single-arm, phase II, multicenter clinical trial, which aimed to target osteosarcomas with pembrolizumab in combination with MC (NCT02406781) [33].

The study included 17 patients, 15 of which were assessable for the primary efficacy endpoint. In order to be included, patients should present with metastatic or unresectable tumor the last six months before entering the study. The primary endpoint was dual-pointed at non-progression and OR at six months. PR was seen only in one patient. Two patients had stable disease (SD). The non-progression rate was not met, reaching only 13.3%. The median PFS was 1.4 months and the median OS was 5.6 months. The most frequent AEs were fatigue and anemia. Fourteen patients had available tissue samples for PD-L1 expression analysis. Two samples showed PD-L1 ≥ 1% in the tumor cells and 1 sample in the immune cells. Four patients showed tumor shrinkage but no one of them expressed any positivity for PD-L1. The authors concluded that this combination of anti-PD1 with MC had insignificant activity in advanced osteosarcoma.

The same investigational team examined the effect of pembrolizumab combined with CM in patients with advanced soft tissue sarcomas (NCT02406781) [34]. The primary endpoints were the same as in the PEMBROSARC study. The study had four arms according to the histological type of the tumor: LMS, UPS, other histological types, and gastrointestinal stromal tumors (GIST). Fifty patients were assessable for the primary efficacy endpoint. The AEs were similar as reported for the PEMBROSARC study. Three patients were progression free at 6 months, while 31 showed disease progression and 16 SD. One patient showed OR. The six-months non-progression was 0%, 0%, 14.3%, and 11.1% for each category, respectively. The median PFS was 1.4 months for each cohort. The median OS was 9.2, 5.6, and 7.1 for the first three categories, respectively, but was not reached for the GIST patients. PD-L1 ≥ 1% was observed on the tumor cells in 12% and on the immune cells in 40% of cases. Interestingly, immune cells were positive in 64% of UPS cases. Only one patient with PR demonstrated a PD-L1 ≥ 10% on immune cells.

The investigators also examined the composition of the tumor microenvironment (TME) and found that the tumors had a high proportion of CD163^+^ macrophages, associated with M2 phenotype known to play a role in immune suppression. This composition ranged from 31% in the LMS arm and reached up to 73% in the UPS arm. In addition, the tumor-associated CD163^+^ macrophages expressed indoleamine-pyrrole 2,3-dioxygenase (IDO1), reaching again 73% in UPSs. Overall, PD-L1 expression in immune cells was significantly positively associated with CD8^+^ cell density and IDO expression. The authors concluded that the M2/IDO suppressor pathway present in most of the investigated sarcomas might play an important role in the resistance to the therapy.

Doxorubicin, a chemotherapeutic agent, alone or in combination with other chemotherapeutic agents, constitutes standard first line systemic treatment for advanced sarcomas [35]. A nonrandomized study addressed the combination of doxorubicin with pembrolizumab in advanced, anthracycline naïve sarcoma patients (NCT02888665) [36]. The study consisted of two phases; in phase I dose-escalation of doxorubicin was examined, starting at 45 mg/m^2^ and increasing up to 75 mg/m^2^ which is the standard treatment dose for sarcomas. A 75 mg/m^2^ dose was well tolerated with no AEs higher than grade 3. In the following phase II patients received 200 mg pembrolizumab and 75 mg/m^2^ doxorubicin every three weeks for up to seven cycles. Thereafter, patients could continue with pembrolizumab as single agent for up to two years. Thirty-seven patients were included in the study, with LMS being the most frequent tumor (11 cases). The primary endpoint was ORR in 15% of the patients, which was not reached, ending in 13% for phase II. On the other hand, encouraging results were found for OS and PFS, being 27.6 and 8.1 months, respectively. PFS at 12 months was 27%. In particular, three out of four patients with UPS and two out of four patients with DDLPS had durable PR, and three out of four patients with chondrosarcoma had tumor regression. PD-L1 expression was very low in almost 70% of the evaluated samples. Of note, a strong association between tumor-infiltrating lymphocytes (TILs) and inferior PFS was seen. Nevertheless, the study did not extensively analyze the composition of the TME.

### 3.4. Immune Checkpoint Inhibitors Combined with Molecular Targeted Therapy

The synergistic effect of targeted therapy and immunotherapy assumes that targeted therapy can have an immunomodulatory effect that increases clinical responses [37].

The single arm phase II trial aimed to combine axitinib, a selective TKI, and pembrolizumab in patients with advanced or metastatic sarcomas (NCT02636725) [38]. Thirty-three patients were enrolled; 36% had ASPS, while the remaining 64% were divided into several histological sarcoma subtypes. Patients received 5 mg axitinib two times a day continuously and 200 mg pembrolizumab every three weeks for cycles of six weeks for up to two years. The investigators applied an intrapatient dose-escalation and de-escalation of axitinib ranging from 2 mg to 10 mg twice daily per cycle. The primary endpoint was PFS at three months. Other endpoints were the rate of participants achieving OR, clinical benefit, OS and the safety and toxicity profile of the drugs.

The three-months PFS rate for all patients that received therapy was 65.6%, but most patients ultimately progressed. Most patients with PR had ASPS. The investigators analyzed ASPS and non-ASPS patients separately for the PFS, OS, and OR. Notably, in non-ASPS the median PFS was similar and the OR was only slightly higher compared to that of patients receiving other types of monotherapy, including axitinib, posing the dilemma whether the addition of an ICI has any value. Still, the six-month PFS is favorable for patients treated with the combined therapy and the investigators concluded that this may be due to the delayed anti-tumor effect of ICIs. The median PFS of the ASPS population is not favorable compared to monotherapy with other broad spectrum TKIs, as reported in literature [39,40,41]. On the other hand, the proportion of patients achieving an OR exceeded the highest previously reported OR of any given monotherapy [39].

All patients with ASPS showed tumor positivity for PD-L1 and a high TIL score, nevertheless, this could not be correlated with a PFS longer than six months nor with a PR.

The most frequent AEs were fatigue and thyroid disorders. Grade 3 and 4 AEs were autoimmune toxic effects, diarrhea, and liver disfunction but the authors concluded that the toxic effects are acceptable and those can be brought under control by axitinib dose escalation.

An investigational team of the University of Miami Miller School of Medicine identified seven patients with angiosarcoma within their institute which were treated with ICIs as monotherapy or combination therapy with other ICIs or TKIs in the context of a clinical trial or off label; after gathering all the available information of each patient, they performed a retrospective study [42]. One of those patients was from the previous described clinical trial that used axitinib and pembrolizumab (NCT02636725) [38]. Five patients showed PR at 12 weeks and 1 showed a CR as best overall response. This patient was treated with anti-CTLA-4 and got different kinds of chemotherapeutic agents in the past. Tissue material gathered from this patient 12 days after the first dose of anti-CTLA-4 revealed that the TME consisted mainly of central memory CD4^+^ and CD8^+^ T-cells, underlining the importance of these cells in the patient’s durable response. The tumor also expressed many novel gene fusions and cancer-testis antigens, which can serve as neoantigens and induce immune response but had a low TMB. The investigators hypothesized that in particular cutaneous angiosarcomas display a comparable mutational signature to ultraviolet-induced skin cancer, such as melanoma, which generally responds well to ICI therapy.

The single-arm, phase II trial (NCT03359018) [43] investigated the synergistic effect of TKIs in combination with ICIs in chemotherapy refractory osteosarcomas. Studies have shown that expression of PD-L1 associates significantly with the presence of T-cells and dendritic cells in the tumor, but also with a poorer five-year event free survival in patients with osteosarcoma [44]. The researches combined 500 mg daily of apatinib, a TKI against vascular endothelial growth factor receptor-2 [45], with 250 mg intravenously carmelizumab given once every two weeks, until disease progression or unacceptable toxicity. The primary endpoint of the study was PFS and the clinical benefit rate (CBR) at six months. Forty-one patients were included. The median PFS was 6.2 months, with a CBR of 30.2%. For the patients with available pretreatment tissue sample for PD-L1 expression, investigators reported a statistically significant PFS in case of PD-L1 ≥ 5%. However, the study did not reach its primary endpoint with a 6-months PFS of 50.9%, which was much lower than the prespecified target of 60%. Moreover, compared to apatinib monotherapy in advanced osteosarcoma [46], the combination treatment did not show superiority. The AEs were in general consistent with the safety profile of the TKIs.

Gastrointestinal stromal tumors (GISTs) nearly always carry activating mutations of c-KIT, a proto-oncogene or platelet-derived growth factor receptor-α gene, giving ground to treatment with TKIs. A phase Ib study of the TKI dasatinib combined with ipilimumab was administrated in 20 GIST and 8 non-GIST sarcoma patients with advanced disease (NCT01643278) [47]. The primary objective was to evaluate the safety profile and to identify the maximum tolerable dose (MTD) of the combination therapy. A dose of 70 mg/day dasatinib and 10 mg/kg ipilimumab was well tolerated with gastric hemorrhage and anemia as the worst grade 3 AEs. After dose escalation of both therapeutic agents, MTD was 140 mg/day dasatinib and 3 mg/kg ipilimumab. The median PFS was 2.8 months and the median OS 13.5 months. No PR or CR was noted. Interestingly, comparing a pre- and two consecutive post-treatment biopsies collected from four patients, IDO was suppressed at the second post-treatment biopsy in a GIST-patient with stable disease for 19 weeks. As such, the investigators suggested IDO suppression might play a role in anti-tumor activity in GISTs. However, this study could not provide convincing evidence for a synergistic effect of dasatinib with ipilimumab for the treatment of GISTs.

### 3.5. Immune Checkpoint Inhibitors Combined with Oncolytic Virus

The effectiveness of talimogene laherparepvec (T-VEC) in combination with pembrolizumab was investigated in patients with advanced or metastatic sarcoma, who had received at least one prior standard therapy line (NCT03069378) [48]. T-VEC is a genetically engineered herpes virus and generates a systemic anti-tumor immune response [49]. Both drugs were administrated every 3 weeks for up to 12 months. This open-label, phase II trial aimed to investigate the ORR at 24 weeks determined by the RECIST version 1.1. Twenty patients with various histological subtypes of sarcoma were included. The primary endpoint was met with an ORR of 30% at 24 weeks. The median time to response was 14.4 weeks while the median duration of response was 56.1 weeks. Epithelioid sarcomas, cutaneous angiosarcomas and unclassified sarcomas were among the histologic types that showed response. Overall, the AEs were not severe and combination of T-VEC with pembrolizumab was well tolerated. Tumor tissue samples from 11 patients were investigated to identify prognostic markers. TIL-score was higher in the responders compared to the refractory group. All responders had CD3^+^/CD8^+^ aggregates at the periphery of the tumor while non-responders did not display this phenotype. In six patients with available pre- and post-treatment tissue, negative PD-L1 expression in pretreatment samples turned positive in post-treatment samples. Compared to the ORR of neoadjuvant chemotherapy in sarcomas that ranges from 16% to 28% [50,51,52], the investigators concluded that T-VEC in combination with pembrolizumab may have a role in treatment of specific histological sarcoma subtypes.

## 4. Discussion

Soft tissue and bone sarcomas is a rare and very heterogenous group of tumors with many subtypes for which diagnosis and treatment remains a very challenging task. On top of that, the treatment choices are limited, and the prognosis of metastatic sarcomas remains poor.

Checkpoint inhibitors have drawn a lot of attention in recent years because of their promising response rates and their durable effects [53]. Nevertheless, ICIs are not a standard treatment choice for sarcomas. Very little data also emerges from the published clinical trials. UPS, DDLPS, and ASPS seem to be good candidates, but it is generally unclear whether ICB, as mono- or combination therapy, is an appropriate treatment for all types of sarcomas.

The possible predictive factors that may play a role in sarcomas’ response to ICB still remain undetermined and require further investigation, as we will discuss.

Some sarcomas express PD-L1. The reported expression varies among different studies. The SARC028 clinical trial (NCT02301039) [18] showed PD-L1 expression in UPS cells, corresponding with an OR after treatment with ICI monotherapy. Moreover, apatinib combined with pembrolizumab in osteosarcomas (NCT03359018) [43] showed that tumors with PD-L1 ≥ 5% correlated significantly with PFS. On the other hand, axitinib in combination with pembrolizumab (NCT02636725) [38] showed PD-L1 expression in all patients with ASPS, but no correlation with PFS for more than 6 months. Furthermore, the study with geptanolimab in patients with advanced ASPS (NCT03623581) [21] demonstrated that almost 1/3 of the tissue samples were positive for PD-L1, but there was no correlation with response. The study with T-VEC and pembrolizumab (NCT03069378) [48] showed expression change of PD-L1 from negative in pre-treatment samples to positive in post-treatment samples from the same patients. A recent study has shown that the proportion of PD-L1 positive osteosarcomas was higher in metastatic than in primary samples [54], emphasizing the ability of the tumor to adapt in order to escape immune response. The remaining trials discussed in this review showed either low PD-L1 IHC expression or no correlation with response. IHC may not represent the actual status of the PD-L1 expression due to the tumor heterogeneity [55]. Moreover, at present, the prognostic and predictive significance of PD-L1 expression in sarcomas is largely unknown. This, together with the variability of expression between the different histological subtypes, poses a challenge in the use of PD-L1 expression as a single predictive biomarker. All these data make it clear that PD-L1 expression in sarcomas disserves to be studied as a separate predictive factor within separate homogenous subtypes of sarcoma.

In addition to PD-L1 expression, the presence of tumor microenvironmental factors also appears to be largely responsible for the response to ICI therapy. Tumors may display a “hot” or a “cold” inflammation signature [56]. Hot tumors are T-cell infiltrated and show a strong immune response to eradicate the tumor. In the SARC028 study (NCT02301039) [18] an inflamed phenotype has been observed in undifferentiated sarcoma which could explain the clinical activity of pembrolizumab. The presence of a strong immune cell infiltrate also suggests activation of immune-suppression pathways that can be targeted [56]. Sarcomas with a complex karyotype such us DDLPS, LMS, and UPS can display an inflamed phenotype and this on its own has been highly associated with clinical response [57]. Moreover, PD-L1 expression has been correlated with T-cell infiltration in UPS [58,59,60]. On the other hand, “cold” tumors have an exhausted or a desert T-cell phenotype. In this regard, activation of the immune response may be the primary scope for tumor elimination. Specific biological mechanisms, such as activation of β-catenin seem to be responsible for T-cell exhaustion and resistance to anti-PD1 therapy [56,61]. Desmoid tumors (DT) are known for showing mutations of the CTNNB1 gene, resulting in activation of the β-catenin pathway. A recent study on DT demonstrated that the tumors have a strong immune infiltrate at the periphery but not within the tumor and do not show a PD-L1 driven immune suppression [62]. Patients with activation of β-catenin are hence very unlikely to benefit from ICI therapy, and β-catenin could be a potential negative predictor biomarker candidate. In melanoma patients it is demonstrated that T cell-mediated cell death can be inhibited by loss of phosphatase and tensin homolog (PTEN) protein [63]. Loss of PTEN is also documented in LMSs and osteosarcomas [64] and this could be a potential mechanism of resistance to ICI therapy. Moreover, limited response of LMS to ICB is confirmed by the studies SARC028 (NCT02301039) [18] and NCT02428192 [20]. Given that PI3K-AKT-mTOR pathway has been proven to be dysregulated via several genetic mechanisms, among them mutations of the PTEN [65], combination therapy of ICB with PI3K-AKT pathway inhibitors may have a role in the treatment of certain sarcomas [63,66,67]. In general, the inflammation signature of sarcomas is not yet clearly described.

In addition to the inflammation signature, also the composition of the TME is of great importance. The first line of defense against the tumor are the cytotoxic CD8^+^ T-cells which presence has been positively correlated with prognosis in different tumor types [68]. However, continuous activation of this defense mechanism can lead to an exhausted T-cell phenotype. On the other hand, the upregulation of T-regs may induce immunologic tolerance [69]. The study of geptanolimab (NCT03623581) [21] demonstrated increased CD4^+^ T-cells in non-responders, whereas CD4^+^ T-cells decreased after treatment in patients with a response. This suggests that CD4^+^ cells might be T-regs and that geptanolimab plays an immunomodulatory role in patients with advanced ASPS. Another important cell type in the TME are the tumor associated macrophages (TAMs). There are two major types of TAMs of which the type 2 (M2) has been described to act in the suppression of the TME and progression of disease [70]. High numbers of TAMs correlate with tumor progression and metastases and have been associated with poor prognosis in gynecological LMS [71,72]. The presence of M2/IDO suppressor pathway in sarcomas might lead to resistance in ICIs, according to the PEMBROSARC study (NCT02406781) [34]. Taken together, it is clear that composition of the TME is dynamic, emphasizing that the response to immunotherapy can be altered. The important question that arises here is how we can monitor such a dynamic change of the TME.

Even if the tumor expresses PD-L1 and the TME seems potent to eliminate the tumor, there are many other factors that can sabotage this process. One of them concerns tumor recognition by the immune system. In order for the immune cells to attack the tumor, the major histocompatibility complex (MHC) has to present neoantigens on the tumor cell surface. As such, loss of MHC could be a possible reason on why tumor cells are not recognized by cytotoxic T-cells. Osteosarcomas can demonstrate variable expression of PD-L1, which could be a potential target, however they also show loss of MHC class I protein indicating immune escape [73].

Secretory factors seem also to interfere with anti-tumor host immunity. Among them, interferon-gamma (IFN-γ) is recently described as an anti-tumor cytokine with an essential role in the polarization of T-helper 1 cells and activation of CD8^+^ cytotoxic T-cells. However, this activation has a negative effect, as the interferon released by those cells can induce expression of PD-L1 by the tumor cells, ultimately stimulating escape from the immune response [74]. A study of Hyuang Kyu Park et al. [75] on sarcomas presented that treatment with IFN-γ increased PD-L1 mRNA levels in different types of sarcoma cell lines. Hence, combination of IFN-γ with anti-PD-L1 agents is a therapeutic possibility for sarcomas that needs further investigation.

The molecular status of tumors gained great interest in the field of immunology the past years. TMB refers to the total number of somatic mutations on coding areas of the tumor genome per megabase. The number of mutations varies among different tumor types. Tumor specific mutations may give rise to neoantigens which can be targeted by T-cells [76]. Statistically, the higher the number of neoantigens, the greater the response to treatment. Although TMB is a promising prognostic biomarker for response to immunotherapy, it does not represent direct evidence of immunogenicity and does not accurately predict the dynamic immune response [77]. Most of the clinical trials discussed in this review did not investigate the role of TMB in sarcomas. The survival benefit of axitinib when combined with pembrolizumab in metastatic sarcomas showed in the NCT02636725 study [38] could not be explained by the high percentage of PD-L1^+^ tumor cells nor by the high TIL scores. Although those tumors lack a high TMB, neoantigens arising from the ASPL-TFE3 fusion with which they are known, could have an immunogenic function [78]. Moreover, the study from the University of Miami [42], found that angiosarcoma patients that showed CR after ICI monotherapy treatment had a very low TMB, but showed novel protein fusions and cancer-testis antigens. On the other hand, all ASPS investigated tissue samples in the study with geptanolimab (NCT03623581) [21], had a very low TMB. Although not all fusions are immunogenic, it has recently been shown that specific fusion-derived neoantigens elicit a cytotoxic T-response including tumors with low TMB. Hence, the possible immunogenicity depends on the expressed fusion protein [79].

A specific type of high TMB tumors demonstrate MSI that induce a hypermutated phenotype. Those tumors generate numerous neoantigens and are highly sensitive to ICI therapy regardless of the tissue of origin [80]. The FDA has granted accelerated approval to pembrolizumab for pediatric and adult patients with MSI-high or mismatch repair-deficient solid tumors [81]. This is the first time the agency has approved a cancer treatment based on a common biomarker rather than an organ-based approach. In this review, only the study with geptanolimab in advanced ASPSs (NCT03623581) [21] investigated tissue samples for MSI, but all tumors were microsatellite stable. Nevertheless, this biomarker seems to play a pivotal role in selecting patients for immunotherapy and further investigation in the context of sarcoma is therefore needed.

As already mentioned, sarcomas is a very heterogenous group of tumors with different subtypes, many of which represent unique diseases with distinct biology. Generally speaking, to date there are no clearly defined biomarkers that predict clinical response of specific histologic subtypes to ICIs. Moreover, given the rarity of sarcomas, it is very difficult to investigate the different histological subtypes separately and come to definite conclusions. What one could summarize from the clinical trials described in this review is that for some histologic sarcoma types that responded in treatment with ICB, specific characteristics might be predictive to the response. Hence, PD-L1 status seems to correlate with better response of endemic/classic type KS. UPS, DDLPS, and LMS can display a “hot” immune signature, implying that the TME might be a predictive factor for these tumor subtypes. An UV-light gene signature in cutaneous angiosarcomas is mentioned as possible indicator for response to ICI therapy. The ASPL-TFE3 gene fusion displayed in ASPSs might be partly responsible for the general good response in most clinical trials. We think that this field disserves further investigation.

## 5. Conclusions

Immune checkpoint blockade gains substantial ground in the cancer treatment. So far, it has shown controversial results in sarcomas. The clinical trials described in this review do not reach a common conclusion or provide strong evidence for the use of any kind of immunotherapy. There are many different etiological leads that could explain those controversies. IHC expression of PD-L1 in sarcomas does not sufficiently correlate with response to ICI therapy, to be used as biomarker. The potential role of IFN-γ on PD-L1 expression by sarcoma cells deserves further attention. The TME composition and TIL counts are very interesting research items for their predictive power, yet not used in daily practice. Specific mutations seem to have a predictive role, such as mutations of the CTNNBN1 or PTEN genes. TMB started as a promising factor, but nowadays we know that it cannot be used as a unique but rather as a complementary marker. MSI strongly correlates to response to ICI therapy and patients with mismatch repair—deficient tumors can benefit from treatment with pembrolizumab regardless of tumor type. Given this complexity and the interaction of various factors in tumors in general, but especially in sarcomas, we believe that in the future a multicomponent predictive biomarker, that will determine which patients are more likely to benefit from treatment with ICI, will be introduced.

## Figures and Tables

**Table 1 cancers-13-00360-t001:** This table summarizes all the clinical trials and their results in terms of the primary endpoint.

Study	Phase	Medication	Targeted Group	Number of Eligible Patients	Tumor Type	Primary Endpoint	Results According to Primary Endpoint
NCT02301039	II	Pembrolizumab	12 years or older	80	Metastatic or surgically unresectable locally advanced soft tissue and bone sarcoma	OR	17.5% soft tissue 5% bone sarcomas
NCT02304458	I-II	Nivolumab	Children and young adults	85	Relapsed or refractory Rhabdomyosarcoma, Ewing sarcoma, osteosarcoma	Tolerability, systemic exposure, MTD and antitumor activity	3 mg/kg every 2 weeks well tolerated No OR
NCT02428192	II	Nivolumab	Adults	12	Advanced UMLS	ORR	0%
NCT01445379	I	Ipilimumab	Children and adolescence	37	Refractory or recurrent sarcomas (and other solid non-sarcoma tumors)	Tolerance and toxicity	Higher grade irAE with increasing dose Better response in patients with high irAE
NCT03623581	II	Geptanolimab	Adults	31	Unresectable, recurrent, or metastatic ASPS	ORR	37.8%
NCT02595866	I	Pembrolizumab	HIV patients	6	Kaposi Sarcoma	Safety of drug	AE similar to non-HIV patients
NCT02500797	II	Nivolumab ± ipilimumab	Adults	76	Metastatic sarcoma	ORR	5% for monotherapy 16% for combination therapy
NCT02406781 (osteosarcoma study)	II	Pembrolizumab + MC	Adults	15	Osteosarcoma	Non-progression and OR at 6 months	Non-progression: 13.3%
NCT02406781 (STS study)	II	Pembrolizumab + MC	Adults	50	LMS UPS Other sarcoma types GIST	Non-progression and OR at 6 months	Non-progression: 0% for LMS/UPS 14.3% for other sarcoma types 11.1% for GIST OR: one patient (2%)
NCT02888665	I-II	Pembrolizumab + Doxorubicin	Adults	37	Advanced Anthracycline-Naive Sarcoma	ORR	19% for phase I 13% for phase II
NCT02636725	II	Pembrolizumab + Axitinib	16 years and older	33	Advanced or metastatic sarcoma	3-months PFS	65.6%
NCT03359018	II	Carmelizumab + apatinib	11 years and older	41	Advanced osteosarcoma	6-months PFS and CBR	PFS: 50.9% CBR: 30.2%
NCT01643278	Ib	Ipilimumab + dasatinib	Adults	28	Refractory GIST and advanced sarcomas	Safety profile and MTD	MDT: dasatinib 140 mg/day + ipilimumab 3 mg/kg
NCT03069378	II	Pembrolizumab + T-VEC	Adults	20	Locally advanced or metastatic sarcoma	Best ORR at 24 weeks	30%

Abbreviations: AE, adverse event; ASPS, alveolar soft part sarcoma; CBR, clinical benefit rate; irAE, immunotherapy associated adverse event; MC, metronomic cyclophosphamide; MTD, Maximum tolerated dose; LMS, leiomyosarcoma; OR, objective response; ORR, objective response rate; STS, soft tissue sarcoma; UPS, undifferentiated pleiomorphic sarcoma.

**Table 2 cancers-13-00360-t002:** Overview of the predictive biomarkers and their clinical importance investigated in the clinical trials described in this review.

Study	Phase	Drug(s)	Predictive Biomarker	Interesting Findings	Clinical Relevance
NCT02301039	II	Pembrolizumab	PD-L1 on TCs Cut-off ≥ 1%	4% PD-L1^+^, all positive samples were UPS	From positive patients: 1 CR and 1 PR
NCT02304458	I–II	Nivolumab	PD-L1 on TCs Cut-off ≥ 1% TME	Low PD-L1 on TCs PD-L1 expression mostly in macrophages	-
NCT02428192	II	Nivolumab	PD-L1 on TCs and ICs PD-1 on ICs	No results available	-
NCT01445379	I	Ipilimumab	Circulating and activated T-cells after ipilimumab administration	Increase of CD4^+^HLA-DR^+^ T cells	No correlation with irAE
NCT03623581	II	Geptanolimab	PD-L1 on TCs Cut-off CPS ≥ 1 MSI TMB Baseline lymphocyte composition	No difference in response between PD-L1 positive and negative TCs Higher percentage CD4^+^ T cells in non-responders	Baseline % CD4^+^ T-cells was negatively associated with patient response
NCT02595866	I	Pembrolizumab	CD4^+^ T-cell count before and after drug administration	CD4^+^ T-cell counts tended to increase	The increases were not statistically significant
NCT02500797	II	Nivolumab ± ipilimumab	PD-L1 TILs TMB T-cell receptor clonality	No results available (ongoing)	-
NCT02406781 (osteosarcoma study)	II	Pembrolizumab + MC	PD-L1 expression on TCs and ICs Cut-off ≥ 1%	TC positivity in 14.3% IC positivity in 7.1%	No correlation of PD-L1 status and clinical response
NCT02406781 (STS study)	II	Pembrolizumab + MC	PD-L1 expression on TCs and ICs (cut-off ≥ 1%) Correlation of M2 macrophage, CD8^+^ and IDO densities	One patient with PR had PD-L1 ≥ 10%, mild IDO1-positive ICs, a CD68^+^ cell density below the median and a very high CD8^+^ cell density The majority of tumors had M2 macrophage that expresses IDO	M2/IDO pathway possibly important mechanism for primary resistance to PD-1 inhibition
NCT02888665	I–II	Pembrolizumab + Doxorubicin	PD-L1 expression (H-Score/MPS) TILs based on morphology Gene expression profile	Expression of PD-L1 was not associated with PFS or OS TILSs present in 29% No gene was significantly associated with PFS	Presence of TILs associated with inferior PFS
NCT02636725	II	Pembrolizumab + Axitinib	PD-L1 expression Presence of TILs	Investigated ASPS tissue samples showed PD-L1 expression and a high TIL score	No PD-L1 and TIL score correlation with PFS (>6 months) or PR
NCT03359018	II	Carmelizumab + apatinib	PD-L1 expression Cut-off ≥ 5% in TCs	No ORR benefit in PD-L1 positive tumors	Prolonged PFS in patients with PD-L1-expressing tumors
NCT01643278	Ib	Ipilimumab + dasatinib	Levels of IDO before and after therapy	IDO suppression in 1 patient with GIST	IDO suppression may potentially correlate with antitumor efficacy in GIST
NCT03069378	II	Pembrolizumab + T-VEC	PD-L1 in TCs Cut-off ≥ 1% TIL score	55% had a turn from PD-L1^−^ at baseline to PD-L1^+^ after treatment Among the responders, one patient with PD-L1^+^ at baseline and 4/9 with PD-L1^+^ posttreatment had PR	All responded patients had higher TIL score mostly in the form of CD3^+^/CD8^+^ aggregates, at the periphery of the tumor

Abbreviations: CD, cluster of differentiation; CPS, Combined Positive Score; CR, complete response; ICs, immune cells; H-score, “histo” score, semiquantitative immunohistochemical scoring; HLA, human leukocyte antigen; IDO, Indoleamine-pyrrole 2,3-dioxygenase; MPS, modified proportion Score; MSI, microsatellite instability; ORR, objective response rate; PFS, Progression Free Survival; PD1, Programmed Death 1; PD-L1, programmed death-ligand 1; PR, partial response; STS, soft tissue sarcoma; TCs, tumor cells; TIL, tumor-infiltrating lymphocyte; TMB, tumor mutational burden; TME, tumor microenvironment; UPS, undifferentiated pleiomorphic sarcoma.

## Data Availability

Not applicable.

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
