# Peer review of "Immune Checkpoint Inhibitory Therapy in Sarcomas: Is There Light at the End of the Tunnel?"

_cancers, 2021, doi:10.3390/cancers13020360_

Round 1

Reviewer 1 Report

It was my honor to review this paper on the use of the immune checkpoint inhibitors in patients with sarcoma.  I believe the paper is interesting, comprehensive and well-written. I have only minor comments.

Comments

line 23 - " the prognosis of aggressive sarcomas is not proportional to the treatment options that exist nowadays". The meaning of this sentence is unclear and it should be removed or rephrased. 

line 79 "The most extensively blockers used are nivolumab and pembrolizumab targeting PD-1 and atezolizumab and durvalumab targeting PD-L1." I recommend that authors do not comment on how extensive the use of each checkpoint inhibitors and rather divide them into PD1 or PDL1 blockers. It is worth to add avelumab and cemiplimab, and mention that many new ones are in clinical trials

Line 84 - it is worth mentioning that ipilimumab in combination with nivolumab can be used not only in melanoma but also in lung cancer and renal cell carcinoma. 

Line 89 - "While NCI’s 5th category only includes the immune inhibitory molecules, it is important to mention that there also exists other immune modulatory molecules with a stimulatory effect such as ICOS, OX40, 4-1BB,…. " This sentence should be rephrased. The NCI categories describe medications, the rest of the sentence talks about molecules (and not medications). 3 dots should removed from the end of the sentence.

Line 147 - "An OR in 25% of the patients was considered clinically meaningful , and a response in less than 10% as ineffective." This sentence can be replaced by a statement that the trial did not reach the prespecified threshold for a 25% response rate. 

Line 161 - It should be discussed if responses were seen in patients with leiomyosarcoma and synovial sarcoma. 

Line 226 - It is a good place to mention the KAPKEY trial presented at ESMO 2020 that showed a response rate of 70% in patients with KS. ref:  ESMO Virtual Congress 2020, Abstract 1077MO

Lime 253 - "These findings are clinically meaningful when compared to the available treatment options for metastatic sarcoma patients." I believe this sentence is confusing. It is unclear which findings it refers to. It should be clearly stated that nivolumab as a single agent was considered inactive; only the combination treatment showed activity that may justify further studies. 

Line 278 - "... had only a modest activity in advanced osteosarcoma." The term "modest activity" is more of a term to state the lack of activity in a more pleasant way. I would recommend the authors not use it and rather possibly say "insignifcant activity"

Line 436. It is extremely important that the authors state clearly that currently available data do not allow us to say if the level of PDL1 expression is a predictive or prognostic biomarker. For examples the authors state in line 441 "that tumors with PD-L1>5% correlated significantly with PFS." We do not if the higher expression of PDL1 correlates with a "better" tumor biology (prognostic biomarker) or actually reflects the increased benefit from therapy (predictive biomarker). This should be discussed. 

Reviewer 2 Report

This paper of Siozopoulou et al. is a concise and timely review that summarizes the current condition of immune checkpoint inhibitory (ICI) therapy for sarcoma. The paper is well organized and important results of ICI therapy on sarcoma is sufficiently reviewed. I would recommend the manuscript published in Cancers after minor modifications. Followings are my suggestions.

  1. In discussion, the authors describe the immune microenvironment in sarcoma  as a potentially important factor for successful ICI therapy. It has been reported that inflammatory and/or immunogenic responses by themselves are important prognostic marker in some sarcomas such as leiomyosarcoma (e.g., TCGA Research Network, Cell, 2017). I would request the authors to add discussion of this point of view.
  2. The effectiveness of ICI therapy with or without TKIs is repeatedly mentioned in the manuscript. In addition, possible role of neoantigens arising from the ASPL-TFE fusion protein is described. However, 30% of sarcomas express fusion proteins resulted from chromosomal aberrations. Does ASPL-TFE3 possess distinct immunogenic property or other fusion proteins also exhibit immunogenic characteristics? Some discussions are required.
  3. If possible, it would be better to have the number of patients in each clinical trial in Table 1.

Reviewer 3 Report

Dr. Siozopoulou reviewed the reports about immune checkpoint inhibitors for sarcoma. 

  1. There are some important reports published only at conference presentation such  as CTOS or ASCO meeting. The reviewer thinks it is better to mention some of those reports. 
  2. Sarcoma is a very heterogenous disease, so it is helpful for readers to summarize and discuss association between tissue type and  biomarkers or response to immune checkpoint inhibitors. It is also helpful to summarize general status of TMB or MSI in sarcoma. 
  3. 3.5  L395  3T-VEC is a onocolytic virus, not vaccine. 
  4. Discussion L476   T cell-mediated cell death can be inhibited (not induced) by loss or PTEN . 

Round 2

Reviewer 3 Report

Dr. Siozopoulou reviewed the reports about immune checkpoint inhibitors for sarcoma. They responded to the reviewer's comments.